# Simulating atmospheric tracer concentrations for spatially distributed receptors: updates to the Stochastic Time-Inverted Lagrangian Transport model's R interface (STILT-R version 2)

Benjamin Fasoli[1], John C. Lin[1], David R. Bowling[2], Logan Mitchell[1], and Daniel Mendoza[1,3]

[1]Department of Atmospheric Sciences, University of Utah, Salt Lake City, 84112, USA
[2]Department of Biology, University of Utah, Salt Lake City, 84112, USA
[3]Division of Pulmonary Medicine, School of Medicine, University of Utah, Salt Lake City, 84112, USA

*Correspondence to:* Benjamin Fasoli (b.fasoli@utah.edu)

**Abstract.** The Stochastic Time-Inverted Lagrangian Transport (STILT) model is comprised of a compiled Fortran executable that carries out advection and dispersion calculations as well as a higher level code layer for simulation control and user interaction, written in the open source data analysis language R. We introduce modifications to the STILT-R codebase with the aim to improve the model's applicability to fine-scale ($< 1\text{km}$) trace gas measurement studies. The changes facilitate placement of spatially distributed receptors and provide high level methods for single and multi-node parallelism. We present a kernel density estimator to calculate influence footprints and demonstrate improvements over prior methods. Vertical dilution in the hyper near-field is calculated using the Lagrangian decorrelation timescale and vertical turbulence to approximate the effective mixing depth. This framework provides a central source repository to reduce code fragmentation between STILT user groups as well as a systematic, well documented workflow for users. We apply the modified STILT-R to light-rail measurements in Salt Lake City, Utah, United States and discuss how results from our analyses can inform future fine-scale measurement approaches and modeling efforts.

## 1 Introduction

Cities are the source of over 70% of global fossil-fuel carbon dioxide ($CO_2$) emissions (International Energy Agency, 2008; Hoornweg et al., 2012; Gurney et al., 2015), the largest anthropogenic forcing on climate change (Canadell et al., 2007). As governing bodies examine ways to address climate change, urban areas are appropriately a focus for emissions regulation. Atmospheric measurements (Duren and Miller, 2012; McKain et al., 2012) provide a top-down constraint for estimating urban carbon emissions, especially when combined with bottom-up information from fuel consumption statistics, traffic data, and building characteristics that result in highly resolved emission inventories (Gurney et al., 2009, 2012). However, traditional evaluation strategies for estimating $CO_2$ emissions that focus on quantifying regional scale ($10^2$ to $10^3$ km) averages at coarse resolutions are unable to resolve urban areas beyond bulk estimates. Implementing and evaluating effective policies for emissions mitigation requires understanding where, when, and how emissions occur at a within-city scale.

Novel measurement strategies are emerging to help resolve fine-scale within-city trace gas concentrations, such as measurements made from trains, buses, and cars (Apte et al., 2017; Bush et al., 2015; Lee et al., 2017) as well as dense networks of inexpensive sensors (Shusterman et al., 2016; Turner et al., 2016). However, traditional atmospheric modeling tools were not designed for densely located and spatially distributed measurements. Simulating atmospheric transport for multiple locations over time often increases the number of simulations by factors of $10^1$ to $10^3$, necessitating the use of scalable parallel computing to best utilize available hardware and reduce total simulation time. To make use of recent measurement advances, modeling approaches must structure the model framework in ways that enable simulations to execute in parallel, adapt to finer spatial scales, and facilitate simulating atmospheric mixing ratios for locations distributed across space and time.

The link between measured atmospheric mole fractions and upstream surface fluxes is often established using Lagrangian particle dispersion models (LPDMs), popular tools for simulating atmospheric transport and dispersion in the Planetary Boundary Layer (PBL) (Lin, 2013). The LPDMs simulate transport of an ensemble of theoretical particles (representing air parcels) using a combination of mean winds interpolated from meteorological model fields with stochastic fluctuations representing turbulent motions introduced as a Markov process. This approach offers advantages over Eulerian methods by explicitly simulating transport trajectories and better representing atmospheric mixing, turbulent eddies, and convection (Lin, 2013). Particle motion can be simulated either forward in time from an emissions source or backward in time from a location of interest, referred to as the "receptor". The forward configuration is often used to simulate pollutant concentrations downstream from an emission source (Stohl et al., 2005) whereas backward simulations determine the source of observed emissions and quantify surface fluxes (McKain et al., 2012, 2015; Stein et al., 2015). As receptors are often greatly outnumbered by sources, significant computational savings are realized by applying LPDMs in the receptor-oriented configuration (Lin, 2013).

The Stochastic Time-Inverted Lagrangian Transport (STILT) model couples Lagrangian particle dispersion with the mean advection scheme from the Hybrid Single-Particle Lagrangian Integrated Trajectory (HYSPLIT) model (Draxler and Hess, 1998; Stein et al., 2015). STILT simulations are reversible in time (Lin et al., 2003), enable quantitative evaluation of transport error (Lin and Gerbig, 2005), and are closely coupled with the commonly used Weather Research and Forecasting mesoscale meteorological model (Nehrkorn et al., 2010), on which the High Resolution Rapid Refresh (HRRR) model is based (Sun et al., 2014). STILT is most commonly used to follow the backwards time evolution of a particle ensemble and calculate a receptor's footprint, a sensitivity matrix defining the upstream area that contributes to tracer mole fractions observed at the receptor. Footprints can be convolved with emissions inventories and an atmospheric background signal to calculate atmospheric mole fractions at the receptor, which is among the most common applications of the STILT model (Gerbig et al., 2003; Lin et al., 2004; Kort et al., 2008; Macatangay et al., 2008; Miller et al., 2008; McKain et al., 2012; Kort et al., 2013; Mallia et al., 2015; McKain et al., 2015).

This paper discusses limitations within the existing STILT codebase and introduces an updated framework intended to improve the model's applicability to fine-scale spatially distributed measurement approaches. Previous work has defined the near-field domain as extending over $10^2$ - $10^3$ km (Lin et al., 2003). We introduce the hyper near-field (HNF) area, typically covering length scales of 1 - 10 km and time scales of 0.1 - 1 hr, from which surface fluxes are diluted to a fraction of the PBL height and thus more strongly influence the receptor. Parameterizations within the STILT model were originally intended

for regional scales and require refinements to improve source-receptor relationships in the HNF. We also describe a footprint calculation scheme using kernel density estimation, rescaling of the effective mixing depth for fluxes in the HNF, and methods for parallelizing simulations. The value of STILT as a tool for interpreting within-city $CO_2$ mole fractions is shown using an example of data collected on the roof of a train car on Salt Lake City (SLC), Utah's light-rail system. We discuss how results
from our analyses can inform future measurement approaches and modeling efforts.

## 2    Modifications to the STILT model

### 2.1    Software enhancements

The R (R Core Team, 2017) component of the STILT model exists as a group of core functions used to track particle locations, calculate footprints, and apply surface flux grids. User groups have built upon these functions, adding scripts for
common modeling workflows and additional functionality. Key components of the higher level functions remain unpublished and undocumented prior to this paper, including a description of methods used to aggregate the particle ensemble to calculate footprints. Here, we adopt a widely-used collaborative software development platform (GitHub) as a common source code repository that meets the needs of STILT users. This repository is built upon existing advection and dispersion calculations but has restructured and modernized the core functions used to interact with the model (Fig. 1).
A single script (*run_stilt.r*) defines model inputs such as receptor locations and meteorological fields, controls and executes the parallelized model, and outputs footprints. Footprints are saved in a netCDF format consistent with conventions for Climate and Forecast metadata (cfconventions.org), the standard for gridded model datasets by the University Corporation for Atmospheric Research (UCAR). This format is compatible with most popular data analysis software platforms and facilitates analysis of model output. The script *run_stilt.r* serves as the primary STILT interface, interacting with R functions which in
turn call Fortran subroutines for the bulk of calculations and providing a systematic, well documented workflow for users.

### 2.2    Model parallelization

Executing simulations in parallel is essential to leverage the full capability of computing resources. STILT receptors are defined in a table of space $(x, y, z)$ and time $(t)$ coordinates enabling users to fix a receptor in space and model the time evolution of the influence field, distribute receptors across space and capture a snapshot at a single time, or distribute the receptors across
both space and time. Since each STILT simulation is computationally independent, total simulation time can be reduced by distributing batches of simulations between parallel threads (Fig. 2). However, past methods for parallelizing simulations require users to manually define batches of receptors and relevant meteorological inputs in unique initialization scripts and submitting each script as a separate job to the scheduler. While increasing the number of parallel threads decreases the size of each simulation batch, the requirements of the user become more complex.
We formalize methods for automatically distributing batches of receptors across many parallel threads managed by the model rather than the user. Within-node parallelism is achieved through process forking, in which batches of receptors are

allocated across multiple parallel threads on a single machine. Multi-node parallelism is accomplished by interfacing with the Simple Linux Utility for Resource Management (SLURM), an open-source tool that provides the framework for interfacing with clusters of computer nodes (Jette and Grondona, 2003). While SLURM is the only cluster job scheduler that has been implemented to date, the open source code can be modified to run on systems managed by other job schedulers including

TORQUE/OpenPBS, Sun Grid Engine, OpenLava, Load Sharing Facility, or Docker Swarm using methods described by Lang et al. (2017). SLURM allocates computational resources with low overhead and can be used to dispatch job arrays of STILT simulations to multiple nodes. SLURM is used to parallelize *between* nodes and process forking by the modified STILT framework is used to parallelize *within* nodes. Process forking can be used independently to execute parallel simulations on a single machine or combined with SLURM to parallelize simulations within each SLURM node. Provided that memory limits

are not exceeded, these methods enable total simulation time to decrease linearly with available CPU cores.

## 2.3 Hyper Near-Field vertical mixing depth

The influence of surface fluxes on air arriving at the receptor depends upon vertical dilution within the atmospheric column. The STILT model determines the height of the boundary layer $z_{pbl}$ using a modified Richardson number method (Vogelezang and Holtslag, 1996). In the original STILT model, surface fluxes are instantaneously diluted within an effective mixing depth of

$h^* = 0.5 \cdot z_{pbl}$ for which the vertical mixing timescale is comparable to the model timestep for advection (Gerbig et al., 2003). As described by Lin et al. (2003), an atmospheric column of height $h(x,y,t,p)$ is used to relate surface fluxes $F(x,y,t)$ to the mole fraction influence $S(x,y,t,p)$ for each particle $p$ as

$$S(x,y,t,p) = \begin{cases} \frac{F(x,y,t)m_{air}}{h(x,y,t,p)\overline{\rho}(x,y,t,p)} & z \leq h \\ 0 & z > h \end{cases} \quad (1)$$

where $\overline{\rho}$ is the average air density below $h$ and $m_{air}$ is the molar mass of dry air. Thus, particles below $h$ perceive surface fluxes

diluted within an atmospheric column of depth $h$. However, the advective timescale is often too short for complete turbulent mixing of HNF fluxes to $h$ before arrival at the receptor. While the assumption that surface fluxes can instantaneously mix to $h^*$ has been validated within the traditional near-field domain (Gerbig et al., 2003), this method underrepresents the influence of HNF fluxes on the tracer mole fraction arriving at the receptor.

We apply a method of calculating the effective mixing depth in the HNF based on homogeneous turbulence theory, described

by Taylor (1922). Each model time step component $k$ of the HNF mixing depth $h'$ at time $t_k$ will be of the form

$$h'_k(p,t) = z_r + \int_0^{t_k} \sigma_w \sqrt{2T_L \left( t + T_L \left( e^{-\frac{t}{T_L}} - 1 \right) \right)} \, dt \quad (2)$$

where $z_r$ is the height above ground of the receptor, $\sigma_w$ is the standard deviation in vertical velocities encountered by $p$ during the integration timestep, and $T_L$ is the Lagrangian decorrelation timescale. Substitution of $h = \min(h', h^*)$ into Eq.

(1) enhances the mole fraction influence of HNF sources on the receptor (Fig. 3). The timestep at which $h' \approx h^*$ signifies the transition between the HNF and traditional near-field domains. The formulation of $h'$ grows the particle-specific dilution depth relative to local turbulence and enables the extent of the HNF domain to vary depending on receptor location, meteorological conditions, and local topography. For surface based applications, $h'$ grows to $h^*$ over 0.1 - 1 hr, affecting a spatial domain of 1 - 10 km adjacent to the receptor.

## 2.4 Kernel density estimation of footprint field

Computing trajectories of large particle ensembles ($N > 10^4$) is computationally expensive. To lessen the cost of each simulation, footprint fields are often calculated from smaller particle ensembles by applying smoothing methods to compensate for the smaller ensemble size. These smoothing methods are less computationally expensive than calculating trajectories for a larger ensemble but vary in their ability to reproduce the robust footprint field of the large particle ensemble.

Prior to methods described in this section, STILT footprints have been calculated by accumulating the influence of particles over an averaging volume. To lessen grid noise from few particles spread throughout the grid, the spatial extent of the particle ensemble was used to dynamically coarsen the size of the averaging volume by a factor of 2 as the particle cloud spreads, first shown in Gerbig et al. (2003). However, at finer resolutions, this method results in excessive smoothing, removing information calculated by the advection and dispersion routines (Fig. 4).

Here we introduce a kernel density estimator to spatially allocate the influence of particles to the footprint grid and show improvements over the prior method at fine spatial resolutions. This method distributes the influence of each particle using a Gaussian weighted spatial kernel centered over the particle's position. The size and intensity of the spatial kernels are defined by the kernel bandwidth, which is determined at each model time step using elapsed time and total dispersion of the stochastic ensemble as proxies for uncertainty in the locations of individual particles. As model time or total ensemble dispersion increase, the kernel bandwidths increase the amount of smoothing applied to each particle. Dispersion of the particle cloud at each time step is represented using a nondimensionalized standard deviation of particle locations $\sigma_d(t)$ given by

$$\sigma_d(t) = \sqrt{\sigma_x^2(t) + \sigma_y^2(t)} \tag{3}$$

where $\sigma_x^2(t)$ and $\sigma_y^2(t)$ are Euclidean variances in the $x$ and $y$ positions of all particles in the ensemble at time $t$. We find $\sigma_d(t)$ to agree with other metrics of dispersion within the particle ensemble, such as the average pairwise distance ($r^2 > 0.99$), with less computational expense. The bandwidths of the gaussian smoothing kernels are then given by

$$b(t) = f \frac{0.06\sqrt{t \cdot \sigma_d(t)}}{\cos\left(\overline{\phi(t)}\right)} \tag{4}$$

where $t$ is time elapsed in days, $\overline{\phi(t)}$ is mean latitude of the particle ensemble used for approximation for meridional grid convergence, and 0.06 is an empirically derived constant. $f$ defaults to 1 and is provided as a user defined smoothing adjustment to enable manual manipulation of kernel sizing.

We test the new footprint calculation methods against a brute force simulation with an atypically large particle ensemble size ($N = 10^5$) aggregated over a homogeneous grid (Fig. 4). This large simulation is computationally expensive but generates an idealized, physically constrained footprint without smoothing algorithms. The simulation receptor was positioned at a SLC $CO_2$ measurement site on a summertime afternoon and particles were followed backward in time for 24 hours. We then demonstrate differences between the new kernel density estimator and the traditional dynamic grid coarsening footprint calculation methods (Fig. 3) for a typical particle ensemble ($N = 200$) and for an extreme case with atypically few particles ($N = 10$). The effects of varying the smoothing parameter ($f = 1, 2$) are shown and errors are quantified using the difference of calculated grid cells from the idealized brute force case.

For the typical case ($N = 200$ and $f = 1$), the kernel density estimator shows improved agreement with the brute force method (rmse $= 5.60 \cdot 10^{-4}$ppm µmol$^{-1}$m$^2$s) compared to the traditional dynamic grid coarsening method (rmse $= 5.79 \cdot 10^{-4}$ppm µmol$^{-1}$m$^2$s), preserving a Gaussian plume adjacent to the receptor, a clustered area of high influence, and capturing split flow upstream. When the kernel bandwidths are doubled by increasing the smoothing parameter ($f = 2$), the footprint field becomes over-smoothed and becomes less similar with the brute force case (rmse $= 5.66 \cdot 10^{-4}$ppm µmol$^{-1}$m$^2$s). In the extreme case using atypically few particles ($N = 10$), the dynamic grid coarsening method produces a footprint field dominated by noise from individual particles (rmse $= 6.12 \cdot 10^{-4}$ppm µmol$^{-1}$m$^2$s). The kernel density estimator ($f = 1$) improves results but shows fragmentation further from the receptor (rmse $= 5.75 \cdot 10^{-4}$ppm µmol$^{-1}$m$^2$s). In this case, the kernel density estimator smoothing parameter enables users to manually widen the plume reproduced in the footprint. Doubling the smoothing parameter ($f = 2$) improves similarity with the smaller particle ensemble (rmse $= 5.70 \cdot 10^{-4}$ppm µmol$^{-1}$m$^2$s) and demonstrates how users can modify the kernel bandwidths to adapt the model to unique cases. While tracer mole fraction differences between the two footprint calculation methods vary depending upon the locations of footprint differences relative to sources, tracer mole fractions calculated using the kernel density estimator are more similar to the idealized brute force case.

## 3 Evaluation

### 3.1 SLC light-rail measurements

We demonstrate these changes to STILT by comparing $CO_2$ mixing ratios simulated by the STILT model with corresponding measurements on-board an electric SLC light-rail commuter train during July 2015. The Salt Lake Valley (SLV) is a 1,300 km$^2$ area encompassing SLC and its surrounding suburbs, bounded by the Wasatch mountain range to the east, the Oquirrh mountain range to the west, the Traverse mountain range to the south, and the Great Salt Lake to the northwest. A light-rail train is equipped to measure high-frequency (1 Hz) $CO_2$ mole fractions in repeated transects of the SLV using a Los Gatos Research Ultraportable Greenhouse Gas Analyzer. $CO_2$ and $CH_4$ mole fractions are corrected for water vapor dilution and spectrum broadening and are calibrated every hour using a compressed whole air tank with known tracer mole fractions traceable to World Meteorological Organization standards. The light-rail train typically operates between the hours of 05:00-23:00 Local Daylight Time (LDT) and only these hours were used in analyses. For details related to the measurement platform, refer to Mitchell et al. (in review).

The observations generally show higher $CO_2$ mole fractions in SLC's urban center and along the north-south oriented urbanized corridor centered in the SLV (Fig. 7), consistent with urban spatial $CO_2$ gradients observed in previous studies (Idso et al., 2001; Pataki et al., 2007). The lowest mole fractions were observed in the southwest corner of the SLV at the margin of recent suburban developments. At a finer scale, the high-frequency measurements show mole fraction enhancements near busy roads and intersections. Measured mole fractions are also consistently higher along a 3 km section of the light-rail track running along the center of a busy six-lane road.

## 3.2 Surface flux inventories

The Hestia bottom up anthropogenic $CO_2$ emissions inventory characterizes carbon fluxes by estimating emissions at the scale of individual buildings and roadways (Gurney et al., 2012). Hestia is available for a handful of U.S. cities including Indianapolis, Los Angeles, Baltimore/D.C., and SLC. Details pertaining to the Salt Lake County Hestia product are described by Patarasuk et al. (2016). For this simulation, Hestia anthropogenic $CO_2$ fluxes are aggregated hourly to a $0.002°$ grid (Fig. 6). However, the Hestia inventory only encompasses the SLV and requires the use of a larger scale anthropogenic emissions inventory to account for fluxes originating from outside of the SLV.

We apply the $1\text{km} \times 1\text{km}$ ODIAC (Oda et al., 2018) inventory to account for anthropogenic $CO_2$ emissions originating outside of the SLV. ODIAC is a globally available gridded dataset that uses power plant profiles and satellite-observed nightlights to spatially allocate estimates of total anthropogenic $CO_2$ emissions. The gridded ODIAC data are temporally allocated to hourly time steps using methods described by Nassar et al. (2013).

Within the SLV domain where the inventories overlap, Hestia and ODIAC agree on the total anthropogenic emissions to within $1.5\%$ during our study period. However, uncertainties of fluxes applied to our analyses are likely larger since the two inventories allocate fluxes differently in space and time. Further, only Hestia is used to represent the SLV whereas ODIAC is used outside of the SLV to account for regional-scale emissions. Uncertainties in inventory estimates are difficult to quantify in time and space and require a devoted effort within the emission inventory scientific community to propagate uncertainties through underlying assumptions within each inventory (Patarasuk et al., 2016; Lauvaux et al., 2016).

The biological $CO_2$ inventory determines land surface types using the 2011 National Land Cover Database (Homer et al., 2007) and 1 m LIDAR derived discrete land cover classifications across the SLV . The link between land cover classification and $CO_2$ exchange is established using AmeriFlux eddy covariance data that provides a robust estimate of biologic fluxes from different vegetation types (Strong et al., 2011). A lookup table with independent axes for temperature, incoming shortwave radiation, and week of year is used to describe the relationship between land cover classification and biological fluxes (Strong et al., 2011) over a $0.01°$ grid. For details pertaining to the biological flux inventory, refer to Strong et al. (2011).

## 3.3 STILT configuration

STILT receptors are defined by averaging light-rail measurements hourly over a $0.002°$ grid (roughly 200 m at $\phi = 41°$ N). The grid resolution was chosen to pair analyses with the $0.002°$ Hestia inventory and because $0.002°$ corresponds roughly with

the size of a SLC block. This method results in 33,608 unique receptors for the month of July, 2015, necessitating the use of parallel simulations and fine scale footprint calculation included in the modified framework.

Urban development and expansion in the area surrounding SLC is limited by the mountainous topography surrounding the city and the Great Salt Lake which restrict the expansion of the city and suburbs. This confines large anthropogenic and biologic sources into a relatively small area surrounding the SLV and simplifies boundary conditions for SLV-centric modeling efforts. From each receptor, 24 h backward trajectories of 200 particle ensembles were calculated using meteorological fields from the HRRR model, available at an hourly interval with a 3 km grid resolution. On average, particles travel within the model domain for 11 h. Computation of the 33,608 particle trajectories and a single set of footprints completed in 5.5 hours utilizing 80 parallel threads across 5 nodes, each equipped with 64 GB of memory with two 8-core Intel XEON E5-2670 2.6 GHz processors. 6.7% of the simulations were not completed due to short-term outages in the HRRR data product.

We compute footprint fields using the legacy dynamic grid coarsening (LEG) algorithm as well as gaussian kernel density estimation with the HNF vertical mixing depth correction (GWD) and without the HNF vertical mixing depth correction (GND) to illustrate the differences between methods. Further, these three methodologies are applied for two model domains, resulting in six different permutations of footprint fields for each receptor. A fine-scale $0.002°$ grid encompasses the SLV and is used to apply SLV anthropogenic emissions. A $0.01°$ grid covering a larger area of Northern Utah is used to apply biological fluxes and non-SLV anthropogenic emissions.

Footprints are convolved with anthropogenic and biological $CO_2$ fluxes and added to background $CO_2$ mole fractions that are representative of $CO_2$ mole fractions that have not been influenced by urban emissions (Mitchell et al., in press). The background mole fractions are taken from a nearby high elevation measurement site at Hidden Peak at the top of the Snowbird ski resort in the Wasatch Mountains (Stephens et al., 2011). We use a similar approach to prior studies (e.g. McKain et al. (2012)) and focus this analyses on the afternoon and early evening hours (12:00-19:00 LDT) to lessen the influence of boundary layer development, nocturnal stratification of the boundary layer, and shallow turbulence on measured mole fractions that would not be represented in the 3 km resolution of the HRRR meteorological fields.

### 3.4   Results

Observed and simulated mole fractions are averaged by hour of day to generate mean diel cycles, shown in Fig. 5. Observations show elevated mole fractions at night and early morning, decreasing into the afternoon as convective mixing increases (Mitchell et al., in press). All three of the simulated diel cycles derived from the different footprint algorithms systematically underestimate nighttime and early morning mole fractions, consistent with previous studies (Macatangay et al., 2008; McKain et al., 2015; Mallia et al., 2015; Lauvaux et al., 2016). However, during afternoon hours the simulated values track more closely with the observations, with GWD exhibiting closer correspondence than GND and LEG (Fig. 5).

Explained variance over space between the time-averaged modeled and measured concentrations is highest for GWD (Pearson's $r^2 = 0.27$) followed by GND ($r^2 = 0.21$) and lastly LEG ($r^2 = 0.20$). While we have demonstrated the gaussian kernel methods to compare favorably with idealized footprints calculated with brute force particle simulations, this analysis found

modeled concentration differences between GND and LEG fall within the uncertainties in surface flux inventories. As GWD agrees most closely with observations over time and space, we focus on GWD for the remainder of analyses.

Footprints convolved with surface flux inventories (Fig. 6) show measurements made on the light-rail train to be highly sensitive to fluxes in the HNF domain. Spatially averaged model results capture the mole fraction gradient between the urban center and surrounding suburbs (Fig. 7). The lowest modeled mole fractions occurred in the southwest corner of the SLV, in agreement with measurements. The model generally produced mole fraction enhancements ($\Delta CO_2$) for grid cells containing or downwind from major roadways (Fig. 7). However, modeling intersection scale enhancements would require finer grid spacing capable of resolving sub-city-block spatial scales that is not yet feasible given current constraints on inventories, meteorological data, and computing resources. On average, we found the largest contributor to modeled $CO_2$ mole fractions is the SLV anthropogenic fluxes ($\Delta CO_2 = 4.18\text{ppm}$), followed by biological fluxes ($\Delta CO_2 = -0.89\text{ppm}$), and the smallest contribution is from non-SLV anthropogenic fluxes ($\Delta CO_2 = 0.37\text{ppm}$).

Key differences between modeled and measured mole fractions exist near HNF sources at the sub-grid scale (Fig. 7). While the model does capture localized mole fraction enhancements near busy roads and intersections ('I' in Fig. 7), measured mole fractions are systematically higher than corresponding model estimates in these areas. These results indicate that the light-rail measurement platform is sampling emissions prior to mixing with the surrounding air. This is evident along the section of light-rail track that shares the six-lane road with other vehicles ('R' in Fig. 7) on which large discrepancies between measurements and model estimates are regularly observed. By diluting emissions throughout a larger grid cell, the model predicts elevated mole fractions localized within and downwind from cells containing significant sources but does not fully capture the magnitude of enhancement resulting from the close proximity to the emissions source.

To demonstrate the benefits of the $0.002°$ grid resolution, we use the above $0.002°$ grid as well as spatially degraded $0.01°$ and $0.1°$ grid resolutions to convolve footprints and fluxes (Fig. 8) and compare against light-rail measurements. Correlations over space between the time-averaged modeled and measured concentrations are highest for the $0.002°$ grid ($r = 0.52$). We find that degrading the resolution to $0.01°$ still captures the SLV-scale urban-suburban-rural mole fraction gradient but fails to resolve much of the roadway and intersection scale enhancements, resulting in a modest decrease in agreement with measurements ($r = 0.48$). However, degrading the resolution to $0.1°$ prevents the model from resolving much of the spatial mole fraction variation ($r = 0.25$). Further, evaluating the spatially-averaged concentration by hour of day shows improved agreement with measurements among the finer grid resolutions ($0.002°$, $0.01°$) over the more coarse $0.1°$ resolution (Fig. 9). While all three resolutions mimic the temporal pattern in the observed mole fraction enhancements due to the temporal variability assigned to emissions inventories, the finer resolutions better capture the mole fraction enhancements observed by the light-rail train in both time and space.

## 4 Summary and Conclusions

In this paper, we have introduced modifications to the STILT-R code that have improved the spatial averaging of the footprints and the model speed. These changes improve the functionality of the STILT model for applications investigating fine-scale

patterns in urban emissions. We have defined the HNF using the effective vertical mixing depth of surface fluxes arriving at a receptor and shown this formulation to improve model agreement with observations. However, calculating the effective vertical mixing depth using turbulence variables $\sigma_w$ and $T_L$ does not extend well to stable nighttime conditions which remain a difficult problem (Holtslag et al., 2013) and a subject of future work.

Given the importance of footprints in the STILT workflow, a kernel density estimator was applied and shown to improve agreement with an idealized brute force method over prior methods. High level methods for single and multi-node parallelism were introduced in this distribution, significantly reducing total simulation time. We then applied STILT to simulate $CO_2$ mole fractions observed along a light-rail train in SLC, at high resolution and show that the model and observations track one another in terms of average spatial and temporal patterns during the afternoon period. However, key differences remain

between modeled and measured mole fractions at night and in the proximity of HNF sources.

     Results indicate that fine-scale inverse analyses will be sensitive to the proximity of observations to upwind sources. Modeling difficulties arise when emissions within the HNF are distributed throughout a larger grid cell that no longer reflects the close proximity of fluxes to the receptor. Fine-scale measurements and modeling approaches are useful for applications such as quantifying pollution exposure in different neighborhoods or locating large point source emitters. However, fluxes originating

within the HNF domain can often dominate the modeled signal and error. Observation techniques that are strongly influenced by HNF fluxes such as trains and cars can have limited usefulness in larger scale applications for which LPDMs have previously been used, such as bulk flux estimates from urban areas. Measurements striving to quantify emissions or assess the validity of emissions inventories should seek to reduce the influence of HNF sources. Prioritizing measurement placement at the top of tall buildings or towers or at least 0.5 km from large sources such as busy roadways enables natural dilution of emitted species,

reduces direct sampling of emissions, and improves model agreement with observations.

*Code availability.* STILT model source code and documentation can be obtained at https://uataq.github.io/stilt/. Development of the model is ongoing and updates will continue to become available through this repository. The precise version of the STILT-R model source code discussed within this manuscript is preserved at http://doi.org/10.5281/zenodo.1196561. Contributions are welcome and should be submitted via pull request. Issues should be reported to the integrated issue tracking system. Questions should be directed to the author.

*Acknowledgements.* This study was supported by NOAA Climate Program Office's Atmospheric Chemistry, Carbon Cycle, and Climate Program, Grant No. NA14OAR4310178. We are grateful to the STILT development community for the prior work that enabled these developments. We thank Douglas Catharine and Philip Dennison for their work assembling the Utah biological flux inventory used in analyses. The support and resources from the Center for High Performance Computing at the University of Utah are gratefully acknowledged. We thank NOAA's Air Resource Laboratory for the HRRR meteorological data. We are grateful to B. Stephens and NCAR's Regional Atmospheric

Continuous $CO_2$ Network (RACCOON) for the high altitude $CO_2$ measurements used as the atmospheric background signal in analyses.

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

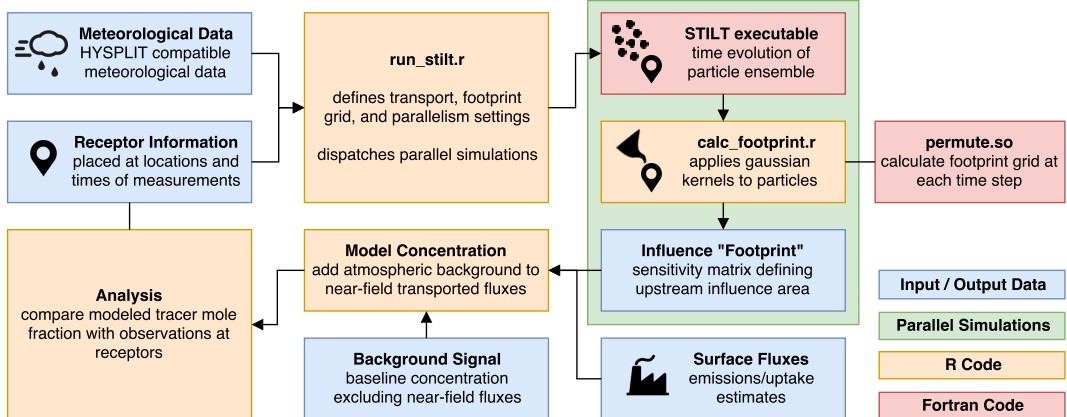

**Figure 1.** STILT workflow to model tracer mole fraction at a receptor. STILT advects particles and calculates the influence footprint for each receptor. Footprints are convolved with surface fluxes and an atmospheric background signal to model the tracer mole fraction.

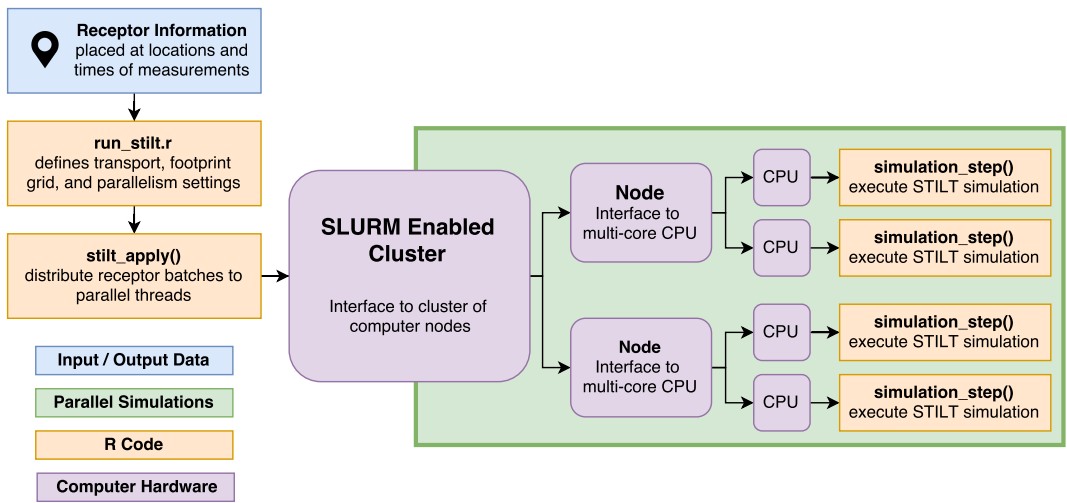

**Figure 2.** Receptor batches are distributed across parallel threads to enable multiple concurrent simulations. Provided memory limits are not exceeded, the total simulation time decreases linearly with the number of CPU cores available.

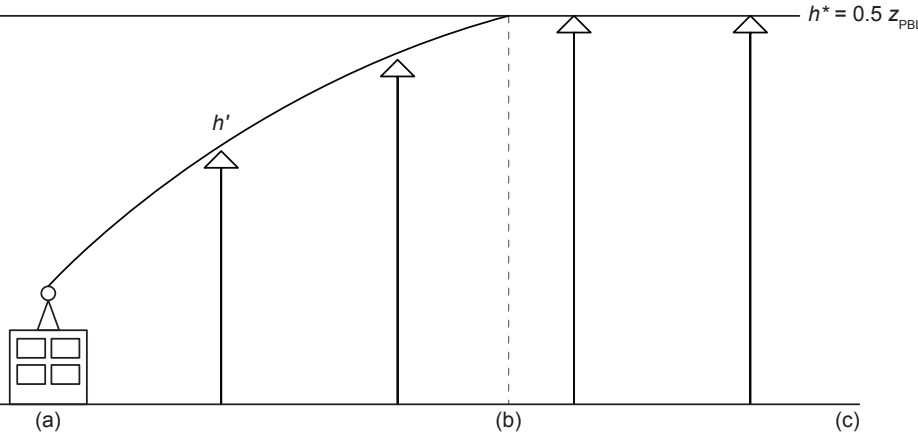

**Figure 3.** Growth of the effective mixing depth. From the receptor (a), surface fluxes are diluted within an atmospheric column depth of $h'$ in the HNF until $h' = h^*$ (b), amplifying the contribution of Hyper Near-Field (HNF) sources and sinks on the receptor. Once $h'$ has reached $h^*$, surface fluxes are diluted to depth $h^*$ until the end of the simulation (c).

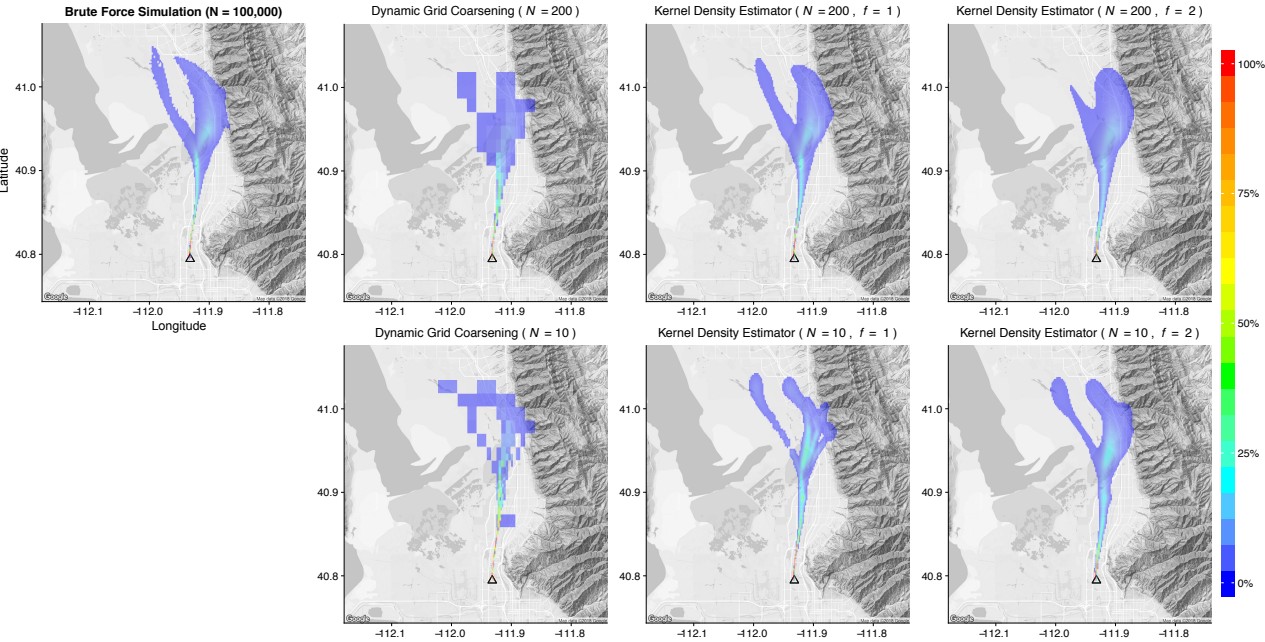

**Figure 4.** Comparison of footprint calculation methods. Simulating a large number of particles ($N = 10^5$) and gridding by location (top left) gives a physically constrained expectation for the footprint. Using subsets of 200 particles (top) and 10 particles (bottom), the kernel density estimator demonstrates considerable improvements over the traditional dynamic grid coarsening. Modifying the kernel bandwidths ($f = 2$) can improve results in uncommon cases, such as the 10 particle ensemble.

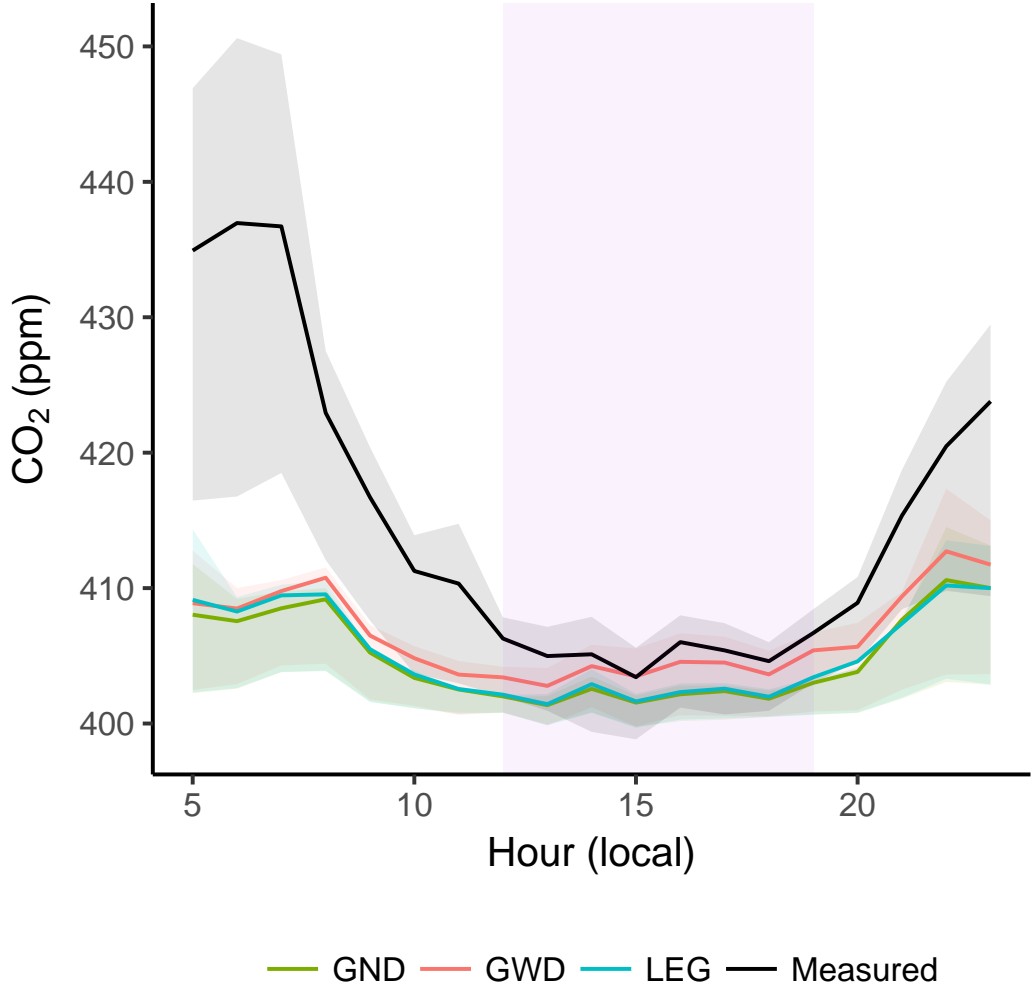

**Figure 5.** Mole fractions of various footprint calculation methods by hour of day during light-rail operating hours from early morning to late evening. Purple shading indicates afternoon hours (05:00-23:00 Local Daylight Time) used for analyses. Solid lines represent the mean and shading represents the interquartile range. Mole fractions modeled using Gaussian kernel calculated footprints with correction for HNF dilution depth (GWD) modeled mole fractions agree most closely with measurements, with underestimation attributed to sub-grid scale sampling of emissions sources.

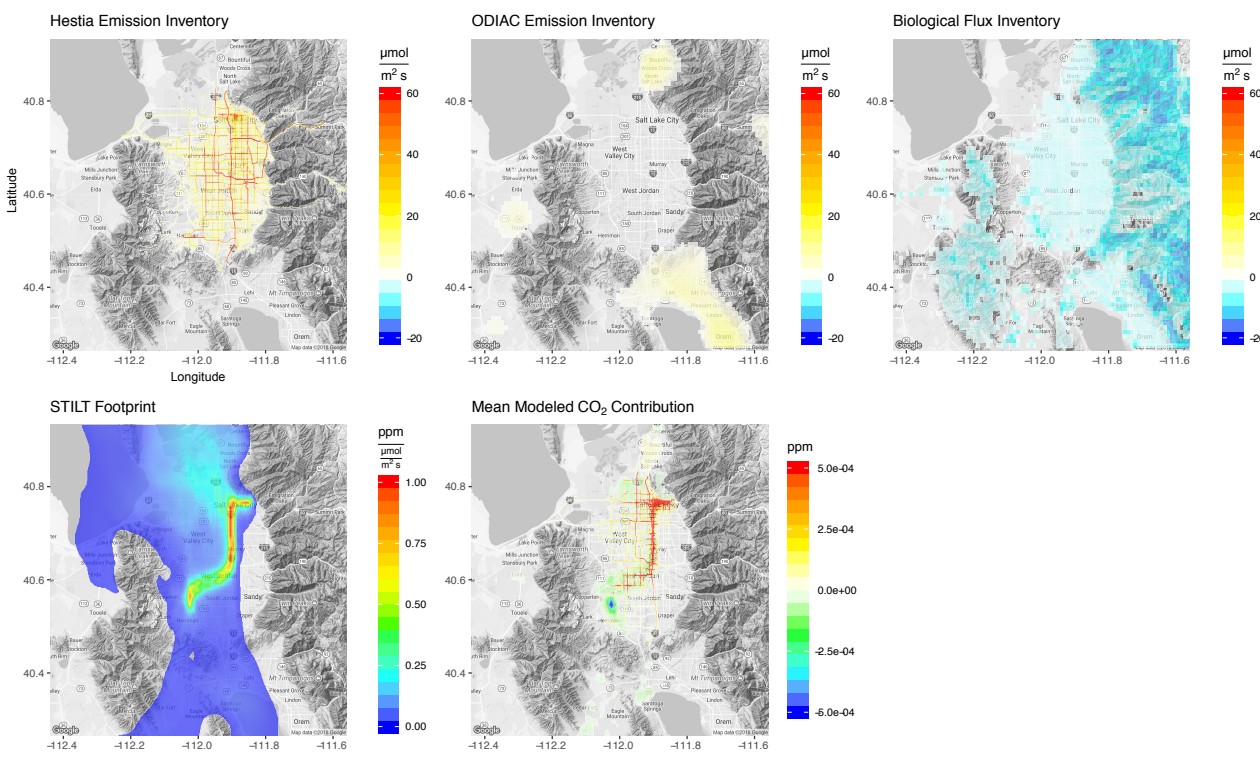

**Figure 6.** July 2015 afternoon Salt Lake Valley (SLV) Hestia-derived and non-SLV ODIAC-derived anthropogenic $CO_2$ emissions, biological fluxes, and average STILT footprint. The anthropogenic and biological flux inventories convolved with the footprints give the contribution of near-field fluxes to measured mole fractions in ppm. The light-rail train is highly sensitive to HNF emissions sources and is strongly influenced by large roadways and agriculture adjacent to the line.

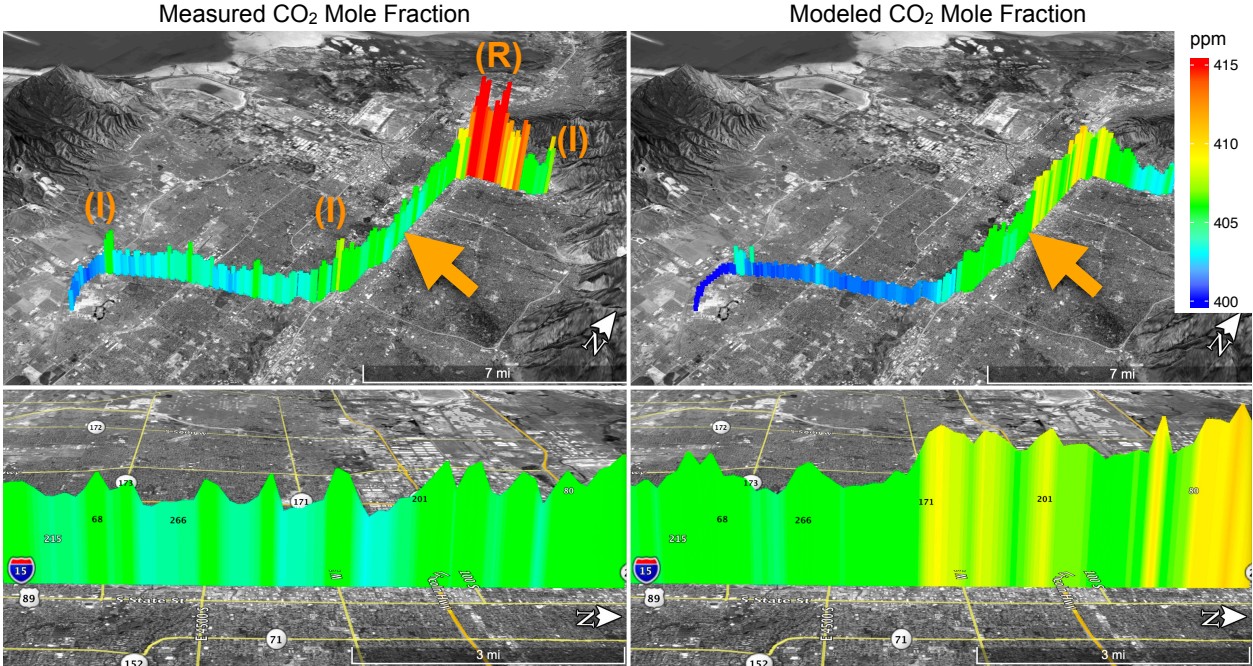

**Figure 7.** Key differences between measured and modeled tracer mole fraction occur near HNF sources, including passing large roadways and intersections (I) and where the light-rail track is shared by other vehicles on the roadway (R). Orange arrow indicates viewpoint of bottom panels. The model captures the overall urban-suburban-rural $CO_2$ mole fraction gradient (top) as well as localized enhancements near grid cells containing large emitters such as busy roads (bottom). Map data: Google, Landsat / Copernicus.

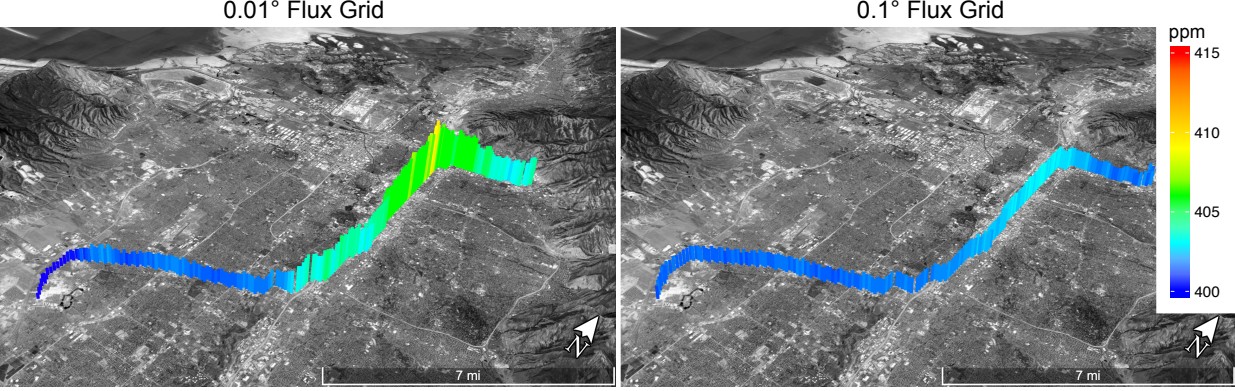

**Figure 8.** Spatially degraded flux and footprint grids to $0.01°$ and $0.1°$ resolutions demonstrates the advantages of the fine-scale, $0.002°$ grid (Fig. 7). While the $0.01°$ resolution (left) retains the $CO_2$ mole fraction enhancements for the SLV-scale urban-suburban-rural gradient, the $0.1°$ resolution (right) fails to resolve the locations and magnitudes of observed $CO_2$ variations. Map data: Google, Landsat / Copernicus.

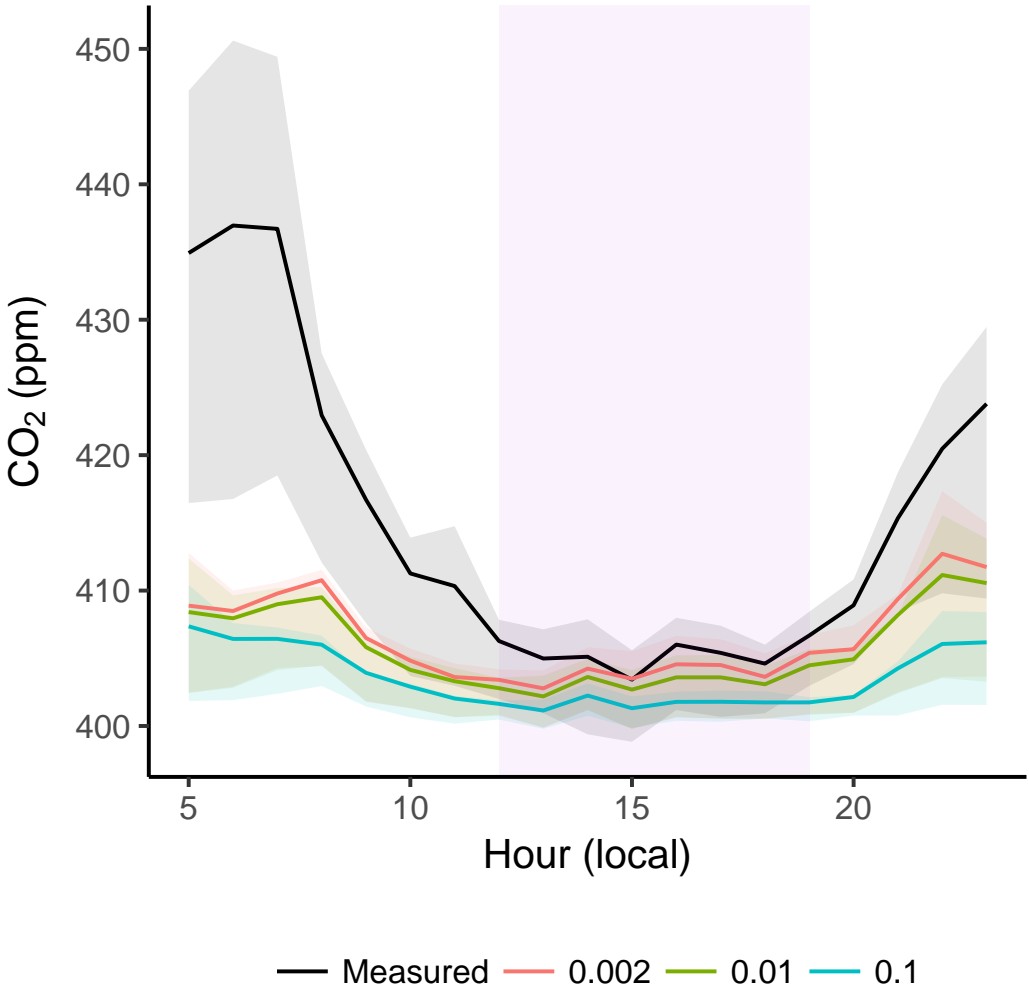

**Figure 9.** Modeled mole fraction at $0.002°$ as well as spatially degraded $0.01°$ and $0.1°$ grid resolutions by hour of day during light-rail operating hours. Purple shading indicates afternoon hours used for analyses. Solid lines represent the mean and shading represents the interquartile range. Mole fractions using finer grid resolutions ($0.002°$, $0.01°$) agree more closely with observations than the coarser $0.1°$ due to the close proximity of the light-rail train to emissions sources.