# Peer review of "Simulating atmospheric tracer concentrations for spatially distributed receptors: updates to the Stochastic Time-Inverted Lagrangian Transport model's R interface (STILT-R version 2)"

_Geoscientific Model Development, 2018_

## Short Comment (SC1) · 12 Mar 2018

The precise version of the code discussed in the manuscript must be made available. The current best practice is for this code to be uploaded to a public repository and a DOI assigned. The DOI should be cited in the manuscript. github is inadequate because it does not readily link to the precise version of the code. However, making github code citable is not difficult; see: https://guides.github.com/activities/citable-code/

---

## Short Comment (SC2) · 13 Mar 2018

Thank you for the recommendation. I have created a release within the Github repository to mark the precise version of the code relevant to the manuscript. This is available at the following URL and Zenodo-registered DOI:

https://github.com/uataq/stilt/releases/tag/v1.0

http://doi.org/10.5281/zenodo.1196561

I will update the text regarding Code Availability accordingly.

---

## Referee Comment (RC1) · Anonymous Referee #1 · 29 Mar 2018

Fasoli et al report on developments made for the Stochastic Time-Inverted Lagrangian Transport Model (STILT). They added high-level functions to make simulations using the R language. They added code to make parallelized simulations. They introduce a new method to deal with near-field emissions which have not yet been homogeneously mixed within the boundary layer. Fasoli et al. further introduce a smoothing

technique to estimate plume surface response functions that is compared to the existing approach. Finally, they show how their developments perform in an experiment in which CO2 measurements have been taken aboard a light-rail in the Salt Lake City metropolitan area.

In general I find the manuscript well written, developments and results are presented in a concise and understandable manner. The manuscript fits the scope and contains enough scientific content to warrant publication in GMD. I have a number of comments that I would like to see addressed before publication, hence I end up with "minor revisions".

My main points of concern are:

1) There is no mentioning of how this work and code repository relates to the original code repository and work at http://stilt-model.org / BGC Jena. None of the co-authors are from Jena or other developers of STILT. I can only assume that this development has been made in accordance and in agreement with the rest of the STILT developers, and that there are no licensing issues. Should be checked and stated explicitly.

2) Model performance is assessed in a very qualitative manner ("looks better") and sometimes overly positive. I suggest authors consider more quantitative assessments, and take a step back before claiming (see below) e.g., that the model represents enhancements in (individual) roadways and intersections.

3) While developing a new method to deal with incompletely mixed sources close to the receptor, the fall back to limiting mixing to the crude 0.5*PBLH formulation. There is no physical basis for that, and I urge the authors to reconsider this artificial limitaion. More on this below.

Specific comments:

p2 l31ff It is unclear to me why 1-10km and 0.1-1 hr spatial and time scales should qualify as "hyper" near-field. Unless you show that "near-field" is a common term

Interactive
comment

that refers to larger spatial or temporal scales I suggest removing the "hyper", as is is hyperbole.

p3 l26ff Can this be used with other queue managers apart from SLURM?

p4 l3ff Again, "hyper near-field" sounds very hyperbole and I suggest renaming it - you are talking about the region in which the well-mixed criterion does not hold.

p4 l7ff Mixing to h = 0.5 * PBLH (p4 l7) is a crude assumption with no physical meaning - if you would wait long enough you would have mixing into 1.0 * PBLH (ignoring en-/detrainment) at the top of the BL.

p4 l15ff You then derive a more complex formulation based on turbulence theory, which you propose to be better. However in the end you use h = min(h', h*), with h* the crude approximation (see above), which effectively stops dilution at 0.5 * PBLH. This seems wrong - why not dilute up to whatever your new formula gives you, maybe cap at 1.0 * PBLH? There is no reason in reality why emissions from the ground should not be mixed further up than half the PBLH.

p4 l21ff: this spatial domain should be variable and strongly dependent on receptor location, topography and meteorolgy - this should be emphasized. In general, sensitivity studies on how this new formulation performs are required.

p4 l21ff: it should be mentioned how (whether?) this method will work with intermittent turbulence and nighttime (stable) conditions (see p7 l22 where you exclude nighttime values for such reasons).

p4 l28: this should be Figure 4.

p5 l8ff: explain better: which particles go into the sigma calculations?

p5 l11: it would be helpful for the reader to see how b enters the two-dimensional Gaussian you are using for density estimation.

p5 l22ff: "improved" is based purely on visual aesthetics ("looks more similar!") - a

quantitative measure would be very beneficial here.

p5 l27ff: "compensating" it might be, but only in the case here - just concede what f is: a fudge factor without physical mean.

p7 l9: "dilution correction" refers to the fudge factor f being set to 2? Explain!

p7 l20ff: You are doing the right thing by ignoring nighttime values, but I suggest you still include the nighttime data in the plots to elucidate the magnitude of this problem - this is something that all model approaches have in common and it helps to remind people that comparing nighttime values is difficult and care needs to be taken.

p8 l7: there is no appreciable "evening rush hour" peak in this figure. Neither does the model "capture" it, as it is too low throughout the day compared to observations. Remove.

p8 l1 and Figure 7 caption: there are no consistent enhancements in modelled $CO_2$ concentrations visible in the bottom right plot that would conincide with the individual intersections shown. I disagree with the statement that the method captures these enhancements. Rephrase and state more careful what you actually can resolve.

p8 l18ff: Careful to make sure that you are not mistaking increasing resolution with the "hyper" near-field approach described earlier - this last section just shows that higher spatial resolution can be beneficial. Might want to rephrase "fine-scale approach".

Figure 6: axis labels missing, should appear at least once for x and y

Figure 5 and 9: plot x axis from 0 to 24, add nighttime values (shade to make clear you don't use them).

Figures 4, 6 - 8: Background maps at least for Figures 7-8 seem to come from Google Earth, are you sure you have the license to use and publish them?

---

## Referee Comment (RC2) · Anonymous Referee #2 · 9 Apr 2018

General Comments:

This work documents the workflow of STILT simulations and presents improved physical processes for fine-scale simulations. I appreciate the authors' efforts in addressing overdue problems for the community, in particular those who use STILT extensively. I hope that the authors continue updating their work through GitHub.

I can easily follow the method and think the paper is relatively well written given the conciseness in length. I have some questions/concerns in the evaluation of the improved method. In current form, the authors do not characterize the errors, in particular in surface emissions. So it is hard to evaluate the results. The model evaluation is a key result in this study, and the authors need to describe how much they know (or pre-scribed) the errors in surface emissions (and others if prescribed) so that we can be sure that the better results from GWD are due to the improved schemes. Please also address the detailed comments below.

Detailed Comments:

L13 - 21: STITL-R should be applicable to other tracer gases, not only CO2. The authors describe CO2 only, which seem to be strange. This is probably because the authors show an evaluation study using CO2, but this CO2 focus is limited.

P2, L21: Need to cite older work about HYSPLIT.

P2, L28 - 29: Need to mention more recent work on city-scale or regional inversion work based on multiple receptors that uses STILT extensively. Literature review here does not represent a full range of the use of the traditional STILT, which I believe is import to for the reader to understand the context, and motivation for the new development.

P3, L6: Need to include the reference for R properly. Not doing so is irresponsible because without R this work is not possible.

P3, L20: For large-scale simulations, the users have applied other types of paralleliza-tions in running STILT, e.g., running multiple jobs (each job may represent one receptor for a give period) at the same time taking advantage of high performance computing. The authors need to briefly mention what the difference between the old method and the one introduced here would be although the method described here seems to be similar to what users have been using. Is there a new concept here?

P3, L27: Not all systems use SLURM although it is popular. Is there an option for a different job scheduling tool?

P4. L4 - 22: In many cases, PBL heights from meteorological models (e.g., WRF) are directly used to represent z_pbl. The authors need to clarify this and describe more on the use of WRF PBL related to equations (1) and (2). For HNF simulations, WRF needs to be run at a similarly fine sale, which is really expensive? If not, what would be the impact on $h = min(h', h^*)$?

P5, L1-2: Reading this, my immediate thought was if this would require more simulation time to estimate the weighted influence. It would be nice to mention the cost.

P6, L32: Should not include a paper in preparation.

P7, L5: 24-h backward in time seems to be too short. How was the upstream boundary condition treated? I see a short description from L17. Boundary conditions are complex due to wind directions. Is the wind consistent from one direction? I would like to see a more description on this.

P7, L30: Please use $r^2$ and state which method was used in calculating r. Pearson's method? How are these $r^2$ values statistically different? The simulations from GWD is distinguishably from a different distribution from the other two so that we have more confidence in GWD? Note that in this evaluation, we want to clearly see better results from GWD. Right?

P8: L1: I think this is probably the most important single statement in this paper. I would like to know how the authors determined the uncertainty in the surface fluxes. Without precise uncertainty characterization, the results are not reliable. What if the inventory is systematically low and GWD overestimated the mole fraction, which could be shown to be closer to the observations than the other two methods? I believe that the authors have considered this point, but I don't see the details here to the level that I can clearly see the outperformance of GWD. Also we need to note that the $r^2$ values are all low and similar to each other.

P8, L6: Please be more quantitative. It is not clear what has been reproduced.

P8, L10 - 15: The simulated mole fractions are a combined result of transport and surface flux emissions. The authors, as mentioned, need to say how much we know about the surface emissions (used here) related to this discrepancy as well as the transport arguably improved from this work.

---

## Author Comment (AC1) · 20 May 2018

**Response to Reviewer 1**

We thank the reviewers for their constructive feedback on our manuscript (https://www.geosci-model-dev-discuss.net/gmd-2018-20/). The reviewers' comments are shown below in *italics* with our responses directly following.

**Anonymous Referee #1**

*Fasoli et al report on developments made for the Stochastic Time-Inverted Lagrangian Transport Model (STILT). They added high-level functions to make simulations using the R language. They added code to make parallelized simulations. They introduce a new method to deal with near-field emissions which have not yet been homogeneously mixed within the boundary layer. Fasoli et al. further introduce a smoothing technique to estimate plume surface response functions that is compared to the existing approach. Finally, they show how their developments perform in an experiment in which CO2 measurements have been taken aboard a light-rail in the Salt Lake City metropolitan area.*

*In general I find the manuscript well written, developments and results are presented in a concise and understandable manner. The manuscript fits the scope and contains enough scientific content to warrant publication in GMD. I have a number of comments that I would like to see addressed before publication, hence I end up with "minor revisions".*

***General comments:***

*1) There is no mentioning of how this work and code repository relates to the original code repository and work at http://stilt-model.org / BGC Jena. None of the co-authors are from Jena or other developers of STILT. I can only assume that this development has been made in accordance and in agreement with the rest of the STILT developers, and that there are no licensing issues. Should be checked and stated explicitly.*

This work is intended to serve as the future replacement for the current "stiltR" wrapper, distributed from the BGC Jena SVN repository. We have been working with the BGC Jena team and the fortran source code remains hosted at BGC Jena. Migrating this wrapper code to GitHub has already enabled significant collaborative development between groups and has led to implementing features beyond those described in this paper.

*2) Model performance is assessed in a very qualitative manner ("looks better") and sometimes overly positive. I suggest authors consider more quantitative assessments, and take a step back before claiming (see below) e.g., that the model represents enhancements in (individual) roadways and intersections.*

Thank you for the suggestion. We have made an effort to improve the assessments of the results to quantify the differences between methods. Please see the specific comments regarding changes to the manuscript.

*3) While developing a new method to deal with incompletely mixed sources close to the receptor, the fall back to limiting mixing to the crude 0.5\*PBLH formulation. There is no physical basis for that, and I urge the authors to reconsider this artificial limitation. More on this below.*

See comment relating to p4 l7ff below.

**Specific comments:**

*p2 l31ff It is unclear to me why 1-10km and 0.1-1 hr spatial and time scales should qualify as "hyper" near-field. Unless you show that "near-field" is a common term that refers to larger spatial or temporal scales I suggest removing the "hyper", as is is hyperbole.*

We thank the reviewer for their comment as because it is important to clarify the naming conventions of domain length scales for readers. We chose the term "hyper near-field" as an extension of the definition of "near-field" in the foundational work of Lin et al 2003, which was "a domain extending over $10^2$-$10^3$km". To clarify this definition in the text, we have added the following statement:

> Previous work has defined the near-field domain as extending over $10^2$ - $10^3$ km (Lin et al., 2003).

*p3 l26ff Can this be used with other queue managers apart from SLURM?*

As of writing, SLURM is the only cluster job scheduler that has been implemented. SLURM is open source and utilized heavily by the high performance computing (HPC) systems at the University of Utah. Due to limited availability of HPC clusters, SLURM is the only job scheduler that has been validated. However, modifications to the project scaffolding described in this manuscript that facilitate parallel computation within single-node and SLURM-scheduled environments opens the doors to other queue managers as well. We encourage future collaboration with users who have access to these job schedulers and would be willing assist with testing development code on their systems. To clarify this in the text, we have added the following statement:

> While SLURM is the only cluster job scheduler that has been implemented to date, the open source code can be modified to run on systems managed by other job schedulers including TORQUE/OpenPBS, Sun Grid Engine, OpenLava, Load Sharing Facility, or Docker Swarm using methods described by Lang et al. (2017).

*p4 l3ff Again, "hyper near-field" sounds very hyperbole and I suggest renaming it - you are talking about the region in which the well-mixed criterion does not hold.*

See comment relating to p2 l31ff above.

*p4 l7ff Mixing to h = 0.5 * PBLH (p4 l7) is a crude assumption with no physical meaning - if you would wait long enough you would have mixing into 1.0 * PBLH (ignoring en- /detrainment) at the top of the BL.*

While we agree that the h = 0.5 PBLH mixing height serves as a crude assumption when used for vertically diluting surface fluxes, it has been extensively validated for the traditional "near-field" domain. This assumption was first introduced by Gerbig et al., 2003 ([https://doi.org/10.1029/2003JD003770](https://doi.org/10.1029/2003JD003770), Section 3.2. Depth of "Surface Layer") who performed a sensitivity study and found that "no significant change in the modeled vegetation signal was found" by varying the fraction of the PBL height considered between 0.1 and 1.0.

The goal of this manuscript is to simplify the model workflow and improve STILT's relevance to the HNF domain. Further, the more complex formulation based on turbulence theory is implemented as an optional feature which can be disabled by the user to replicate past simulations while taking advantage of the other improvements described in this manuscript. With this in mind, we retained the traditional h = 0.5 PBLH for the "near-field" domain for consistency with previous work.

The following text has been added:
> While the assumption that surface fluxes can instantaneously mix to $h^*$ has been validated within the traditional near-field domain (Gerbig et al., 2003), this method underrepresents the influence of HNF fluxes on the tracer mole fraction arriving at the receptor.

*p4 l15ff You then derive a more complex formulation based on turbulence theory, which you propose to be better. However in the end you use h = min(h', h*), with h* the crude approximation (see above), which effectively stops dilution at 0.5 * PBLH. This seems wrong - why not dilute up to whatever your new formula gives you, maybe cap at 1.0 * PBLH? There is no reason in reality why emissions from the ground should not be mixed further up than half the PBLH.*

As you say, the formulation of h = min(h', h*) results in the use of the turbulence theory estimation until the traditional 0.5 PBLH is met. The model timestep at which h' is approximately equal to h* is the transition between the "hyper near-field" and "near-field" domains. Rather than a rigid definition of the "hyper near-field" length scale, setting the dilution depth to min(h', h*) allows the "hyper near-field" spatial domain to adapt to meteorological conditions while enabling a smooth transition between the two methods of vertically diluting surface fluxes.

To clarify this point in the manuscript, the following text has been added:

> The timestep at which h' ≈ h* signifies the transition between the HNF and traditional near-field domains. The formulation of h' grows the particle-specific dilution depth relative to local turbulence and enables the extent of the HNF domain to vary depending on receptor location, meteorological conditions, and local topography.. For surface based applications, h' grows to h* over 0.1 - 1 hr, affecting a spatial domain of 1 - 10 km adjacent to the receptor.

*p4 l21ff: this spatial domain should be variable and strongly dependent on receptor location, topography and meteorology - this should be emphasized. In general, sensitivity studies on how this new formulation performs are required.*

We agree that it the spatial domain of the HNF is highly variable and have updated the text to clarify. See comment relating to p4 l15ff above.

*p4 l21ff: it should be mentioned how (whether?) this method will work with intermittent turbulence and nighttime (stable) conditions (see p7 l22 where you exclude nighttime values for such reasons).*

We agree that we should emphasize that nighttime turbulence is an unsolved problem that is beyond the scope of this manuscript.

The following text was added to ~p9L10:

> We have defined the HNF using the effective vertical mixing depth of surface fluxes arriving at a receptor and shown this formulation to improve model agreement with observations. However, calculating the effective vertical mixing depth using turbulence variables $\sigma_w$ and $T_L$ does not extend well to stable nighttime conditions which remain a difficult problem (Holtslag et al., 2013) and a subject of future work.

*p4 l28: this should be Figure 4.*

Thanks for catching that mistake.

*p5 l8ff: explain better: which particles go into the sigma calculations?*

We have modified the equation notation and text to show that the sigma calculations are derived from the positions of all particles in the ensemble at each model timestep.

To clarify this in the text, we have modified the text following equation 3 to include:

> where $\sigma_x^2$ and $\sigma_y^2$ are Euclidean variances in the x and y positions of all particles in the ensemble at time t.

*p5 l11: it would be helpful for the reader to see how b enters the two-dimensional Gaussian you are using for density estimation.*

While we agree that is important to understand how the kernel bandwidth relates to the smoothing applied, visualizations and discussions regarding bandwidth selection and how it relates to bias-variance optimization can be found in many general descriptions of kernel density estimation.

To help the reader understand what effect the bandwidth has on the model output without requiring an additional visualization, we have added the following text to ~p5L17:

> As model time or total ensemble dispersion increase, the kernel bandwidths increase the amount of smoothing applied to each particle.

*p5 l22ff: "improved" is based purely on visual aesthetics ("looks more similar!") - a quantitative measure would be very beneficial here.*

Agreed. We have performed additional analysis and added a quantitative measure for the difference between the calculation methods and the ideal case.

The following text has been added to ~p6L1:

> The effects of varying the smoothing parameter (f = 1,2) are shown and errors are quantified using the difference of calculated grid cells from the idealized brute force case.
>
> For the typical case (N = 200 and f = 1), the kernel density estimator shows improved agreement with the brute force method (rmse = 5.60 * $10^{-4}$ ppm (umol$^{-1}$ m$^2$ s)) compared to the traditional dynamic grid coarsening (rmse = 5.79 * $10^{-4}$ ppm (umol$^{-1}$ m$^2$ s)), preserving a Gaussian plume adjacent to the receptor, a clustered area of high influence, and capturing split flow upstream. When the kernel bandwidths are doubled by increasing the smoothing parameter (f = 2), the footprint field becomes over-smoothed and becomes less similar with the brute force case (rmse = 5.66 * $10^{-4}$ ppm (umol$^{-1}$ m$^2$ s)). In the extreme case using atypically few particles (N = 10), the dynamic grid coarsening method produces a footprint field dominated by noise from individual particles (rmse = 6.12 * $10^{-4}$ ppm (umol$^{-1}$ m$^2$ s)). The kernel density estimator (f = 1) improves results but shows fragmentation further from the receptor (rmse = 5.75 * $10^{-4}$ ppm (umol$^{-1}$ m$^2$ s)). In this case, the kernel density estimator smoothing parameter enables users to manually widen the plume reproduced in the footprint. Doubling the smoothing parameter (f = 2) improves similarity with the smaller particle ensemble (rmse = 5.70 * $10^{-4}$ ppm (umol$^{-1}$ m$^2$ s)) and demonstrates how users can modify the kernel bandwidths to adapt the model to unique cases. While tracer mole fraction differences between the two footprint calculation methods vary depending upon the locations of footprint differences relative to sources, tracer mole fractions calculated using the kernel density estimator are more similar to the idealized brute force case.

*p5 l27ff: "compensating" it might be, but only in the case here - just concede what f is: a fudge factor without physical mean.*

We have changed the language to highlight the reviewer's comment at ~p6L12:
> In this case, the kernel density estimator smoothing parameter enables users to manually widen the plume reproduced in the footprint. Doubling the smoothing parameter (f = 2) improves similarity with the smaller particle ensemble (rmse = 5.70 * $10^{-4}$ ppm ($umol^{-1}$ $m^2$ s)) and demonstrates how users can modify the kernel bandwidths to adapt the model to unique cases.

*p7 l9: "dilution correction" refers to the fudge factor f being set to 2? Explain!*

We agree that using the terms "dilution correction" and "vertical mixing depth correction" interchangeably was confusing for readers. We have changed the text to "HNF vertical mixing depth" to be consistent with terms used in the methods description (Section 2.3).

The following text has been added to ~p7L14:
> We compute footprint fields using the legacy dynamic grid coarsening (LEG) algorithm as well as gaussian kernel density estimation with the HNF vertical mixing depth correction (GWD) and without the HNF vertical mixing depth correction (GND) to illustrate the differences between methods.

*p7 l20ff: You are doing the right thing by ignoring nighttime values, but I suggest you still include the nighttime data in the plots to elucidate the magnitude of this problem - this is something that all model approaches have in common and it helps to remind people that comparing nighttime values is difficult and care needs to be taken.*

While we agree that it is useful to compare nighttime modeled values between manuscripts, the light-rail measurement platform only operates during specific hours of the day. We have clarified this in the text.

The following text has been added to ~p6L14:
> The light-rail train typically operates between the hours of 05:00-23:00 Local Daylight Time (LDT) and only these hours were used in analyses.

*p8 l7: there is no appreciable "evening rush hour" peak in this figure. Neither does the model "capture" it, as it is too low throughout the day compared to observations. Remove.*

We agree that discussing an "evening rush hour peak" may be inaccurate in this context. The late afternoon increase in modeled $CO_2$ is the result of increased emissions from both anthropogenic inventories as well as meteorological factors. We have removed the text as the reviewer suggested.

*p8 l1 and Figure 7 caption: there are no consistent enhancements in modelled CO2 concentrations visible in the bottom right plot that would conincide with the individual intersections shown. I disagree with the statement that the method captures these enhancements. Rephrase and state more careful what you actually can resolve.*

We agree that it is important to not inflate the improvements and the language describing spatial resolution needs to be more clearly defined.

The following text was added to ~p7L18:

This grid resolution was chosen to pair analyses with the 0.002° Hestia inventory and because 0.002° corresponds roughly with the size of a Salt Lake City block.

The following text was added to ~p8L21:

The model generally produced mole fraction enhancements ($\Delta CO_2$) for grid cells containing or downwind from major roadways (Fig. 7). However, modeling intersection scale enhancements would require finer grid spacing capable of resolving sub-city-block spatial scales that is not yet feasible given current constraints on inventories, meteorological data, and computing resources.

The following text has been added to the Fig. 7 caption:

The model captures the overall urban-suburban-rural $CO_2$ mole fraction gradient (top) as well as localized enhancements near grid cells containing large emitters such as busy roads (bottom).

*p8 l18ff: Careful to make sure that you are not mistaking increasing resolution with the "hyper" near-field approach described earlier - this last section just shows that higher spatial resolution can be beneficial. Might want to rephrase "fine-scale approach".*

We agree that readers may confuse the language with the hyper near-field definition. The text "fine-scale" has been changed to "0.002° grid resolution".

*Figure 6: axis labels missing, should appear at least once for x and y*

We have added the axis labels (Longitude and Latitude) to Figure 6 as recommended.

*Figure 5 and 9: plot x axis from 0 to 24, add nighttime values (shade to make clear you don't use them).*

The hours that are not represented on the x-axis do not contain any data. See comment relating to p7 l20ff above for details.

*Figures 4, 6 - 8: Background maps at least for Figures 7-8 seem to come from Google Earth, are you sure you have the license to use and publish them?*

Google Maps and Google Earth permits use in periodicals (https://www.google.com/permissions/geoguidelines.html#maps-print) with proper attribution. However, it appears that the attributions were cropped out of several of the figures. The content was updated in accordance with Google's attribution guidelines (https://www.google.com/permissions/geoguidelines/attr-guide.html).

---

## Author Comment (AC2) · 20 May 2018

**Response to Reviewer 2**

We thank the reviewers for their constructive feedback on our manuscript (https://www.geosci-model-dev-discuss.net/gmd-2018-20/). The reviewers' comments are shown below in *italics* with our responses directly following.

**Anonymous Referee #2**

*This work documents the workflow of STILT simulations and presents improved physical processes for fine-scale simulations. I appreciate the authors' efforts in addressing overdue problems for the community, in particular those who use STILT extensively. I hope that the authors continue updating their work through GitHub.*

*I can easily follow the method and think the paper is relatively well written given the conciseness in length.*

Thanks for the positive comments. We hope that readers will agree.

*I have some questions/concerns in the evaluation of the improved method. In current form, the authors do not characterize the errors, in particular in surface emissions. So it is hard to evaluate the results. The model evaluation is a key result in this study, and the authors need to describe how much they know (or pre- scribed) the errors in surface emissions (and others if prescribed) so that we can be sure that the better results from GWD are due to the improved schemes.*

We have added a discussion regarding difficulties in estimating uncertainties in emissions inventories. Please see below for details.

***Detailed Comments:***

*L13 - 21: STITL-R should be applicable to other tracer gases, not only CO2. The authors describe CO2 only, which seem to be strange. This is probably because the authors show an evaluation study using CO2, but this CO2 focus is limited.*

STILT's applications certainly exceed only simulating atmospheric CO2. We attempt to describe the use of LPDMs and the STILT model (~p2L9, ~p2L20) using generalized language such as "atmospheric mole fractions", "pollutant concentrations", and model applicability to "observed emissions" and "surface fluxes". We use urban $CO_2$ as the primary motivation for several reasons: urban $CO_2$ cycling is the focus of a large and growing body of scientific literature that this model update will play a prominent role in, it allows for the use of novel $CO_2$ surface flux

inventories purpose-built for the study region (the Hestia model), and it applies well to the case study using the unique data available from the light-rail measurement system.

*P2, L21: Need to cite older work about HYSPLIT.*

We have added a citation for Draxler, R.R., and G.D. Hess, 1998.

*P2, L28 - 29: Need to mention more recent work on city-scale or regional inversion work based on multiple receptors that uses STILT extensively. Literature review here does not represent a full range of the use of the traditional STILT, which I believe is import to for the reader to understand the context, and motivation for the new development.*

We have added citations for McKain et al., 2012 and McKain et al., 2015 describing STILT modeling applications in Salt Lake City and Boston as well as Kort et al., 2013 describing STILT's use to assess measurement network design in Los Angeles.

*P3, L6: Need to include the reference for R properly. Not doing so is irresponsible because without R this work is not possible.*

Thank you for the suggestion. We have added the citation for the R software at ~p3L7.

*P3, L20: For large-scale simulations, the users have applied other types of parallelizations in running STILT, e.g., running multiple jobs (each job may represent one receptor for a give period) at the same time taking advantage of high performance computing. The authors need to briefly mention what the difference between the old method and the one introduced here would be although the method described here seems to be similar to what users have been using. Is there a new concept here?*

We recognize that we did not adequately describe past efforts to run parallel simulations. While the concept of executing batches of receptors across multiple jobs is not new, users have previously had to write and run separate scripts defining the receptors and relevant data inputs for each job which can require significant manual labor or develop their own methods for batch processing receptors. The manuscript formalizes methods for automatically executing the parallel batches of receptors, with receptor batches distributed between the parallel jobs and managed by the code itself rather than the user. The workflow presented, controlled with *run_stilt.r* and with output saved to simulation ID directories, remains the same for serial and parallel execution with only changing the setting for the number of parallel processes.

To clarify this point, the following text has been added to ~p3L28 :

>However, past methods for parallelizing simulations require users to manually define batches of receptors and relevant meteorological inputs in unique initialization scripts and submitting each script as a separate job to the scheduler. While increasing the

number of parallel threads decreases the size of each simulation batch, the requirements of the user become more complex.

We formalize methods for automatically distributing batches of receptors across many parallel threads managed by the model rather than the user.

*P3, L27: Not all systems use SLURM although it is popular. Is there an option for a different job scheduling tool?*

As of writing, SLURM is the only cluster job scheduler that has been implemented. SLURM is open source and utilized heavily by the high performance computing (HPC) systems at the University of Utah. Due to limited availability of HPC clusters, SLURM is the only job scheduler that has been validated. However, modifications to the project scaffolding described in this manuscript that facilitate parallel computation within single-node and SLURM-scheduled environments opens the doors to other queue managers as well. We encourage future collaboration with users who have access to these job schedulers and would be willing assist with testing development code on their systems. To clarify this in the text, we have added the following statement:

> While SLURM is the only cluster job scheduler that has been implemented to date, the open source code can be modified to run on systems managed by other job schedulers including TORQUE/OpenPBS, Sun Grid Engine, OpenLava, Load Sharing Facility, or Docker Swarm using methods described by Lang et al. (2017).

*P4. L4 - 22: In many cases, PBL heights from meteorological models (e.g., WRF) are directly used to represent z_pbl. The authors need to clarify this and describe more on the use of WRF PBL related to equations (1) and (2). For HNF simulations, WRF needs to be run at a similarly fine sale, which is really expensive? If not, what would be the impact on h = min(h',hˆ*)?*

The formulation for the HNF vertical mixing depth adjustment h' is intended to fix systematically low footprints without needing to explicitly resolve $z_{pbl}$ at HNF resolutions. It provides an estimate for the effective mixing depth based on homogeneous turbulence theory without requiring meteorological inputs (e.g. WRF) to be at a scale that explicitly defines the fine variations in PBL height within a city. However, the meteorological data are used outside of the HNF domain to calculate h* using a modified Richardson number method that has been extensively validated for the traditional "near-field" domain.

*P5, L1-2: Reading this, my immediate thought was if this would require more simulation time to estimate the weighted influence. It would be nice to mention the cost.*

Agreed. Calculating the footprint field using smoothing methods involves a cost tradeoff with a larger particle ensemble. While it is almost always less expensive to apply smoothing methods compared to calculating particle trajectories, quantifying the advantage is difficult. The cost to

calculate particle trajectories varies depending on model configuration, meteorological data source, the size of the meteorological domain, and the size of the ensemble while the cost to apply smoothing depends on the method and the spatial and temporal domain of the output footprint.

To clarify this point, the following text has been added to ~p5L1:
> Computing trajectories of large particle ensembles ($N > 10^4$) is computationally expensive. To lessen the cost of each simulation, footprint fields are often calculated from smaller particle ensembles by applying smoothing methods to compensate for the smaller ensemble size. These smoothing methods are less computationally expensive than calculating trajectories for a larger ensemble but vary in their ability to reproduce the robust footprint field of the large particle ensemble.

*P6, L32: Should not include a paper in preparation.*

Agreed. We removed the citation since the manuscript is still in preparation.

*P7, L5: 24-h backward in time seems to be too short. How was the upstream boundary condition treated? I see a short description from L17. Boundary conditions are complex due to wind directions. Is the wind consistent from one direction? I would like to see a more description on this.*

We find that particles exist within the footprint domain for 11 hours on average. The meteorological domain encompasses a larger area than the footprint domain and fluxes from outside of the footprint domain are assumed to be resolved by the background atmospheric signal described at p8L6.

To clarify this point, the following text has been added to ~p7l24:
> Urban development and expansion in the area surrounding SLC is limited by the mountainous topography surrounding the city and the Great Salt Lake which restrict the expansion of the city and suburbs. This confines large anthropogenic and biologic sources into a relatively small area surrounding the SLV and simplifies boundary conditions for SLV-centric modeling efforts. From each receptor, 24 h backward trajectories of 200 particle ensembles were calculated using meteorological fields from the HRRR model, available at an hourly interval with a 3 km grid resolution. On average, particles travel within the model domain for 11 h. Computation of the 33,608 particle trajectories and a single set of footprints completed in 5.5 hours utilizing 80 parallel threads across 5 nodes, each equipped with 64 GB of memory with two 8-core Intel XEON E5-2670 2.6 GHz processors. 6.7% of the simulations were not completed due to short-term outages in the HRRR data product.

*P7, L30: Please use $r^2$ and state which method was used in calculating r. Pearson's method? How are these $r^2$ values statistically different? The simulations from GWD is distinguishably*

*from a different distribution from the other two so that we have more confidence in GWD? Note that in this evaluation, we want to clearly see better results from GWD. Right?*

As recommended, we have modified the text to use $r^2$ instead of r to explain model variance and have clarified that it is based on Pearson's method.

While there is likely no statistical significance in the differences between GND and LEG for this case study, we show that GND agrees better with the physical "ideal" case and may give improvements that depend on the locations of differences between GND and LEG relative to the locations of surface fluxes. With the vertical dilution correction (GWD), the results agree more closely with measurements in both time and space.

*P8: L1: I think this is probably the most important single statement in this paper. I would like to know how the authors determined the uncertainty in the surface fluxes. Without precise uncertainty characterization, the results are not reliable. What if the inventory is systematically low and GWD overestimated the mole fraction, which could be shown to be closer to the observations than the other two methods? I believe that the authors have considered this point, but I don't see the details here to the level that I can clearly see the outperformance of GWD. Also we need to note that the r^2 values are all low and similar to each other.*

We agree that it is important to investigate uncertainty in inventory estimates. While we can show improvements to footprint smoothing algorithms using physically constrained "ideal" cases, uncertainty estimates within emissions inventories remains an unresolved question within the emission inventory scientific community. Developers of the Hestia inventory have documented that "a devoted effort is needed to generate uncertainty and propagate those uncertainties through the Hestia approach to provide an improved understanding of where results are more or less certain in space and time. This remains a high priority for future research" (Patarasuk et al., 2016) and determination of GHG fluxes and uncertainty bounds is one of the primary goals in the ongoing Indianapolis Flux Experiment (http://sites.psu.edu/influx/). Improvements to LPDMs can help future inverse modelling frameworks that would be better equipped to quantify uncertainties in flux inventories.

To further clarify this, we have added a discussion regarding the difficulties in assessing emission inventory uncertainties. Both of the inventories we discussed in the manuscript (Hestia and ODIAC) agree on the total emissions within the SLV domain which is evidence one inventory is not systematically lower than the other. However, mapping uncertainty to a moving receptor using two emissions inventories that encompass different spatial domains and allocate fluxes using different methods in time and space is a difficult question that requires more tools and analysis than are available in our present manuscript and should be the focus of future work.

The following text has been added to ~p7L20:

Within the SLV domain where the inventories overlap, Hestia and ODIAC agree on the total anthropogenic emissions to within 1.5% during our study period. However, uncertainties of fluxes applied to our analyses are likely larger since the two inventories allocate fluxes differently in space and time. Further, only Hestia is used to represent the SLV whereas ODIAC is used outside of the SLV to account for regional-scale emissions. Uncertainties in inventory estimates are difficult to quantify in time and space and require a devoted effort within the emission inventory scientific community to propagate uncertainties through underlying assumptions within each inventory (Patarasuk et al., 2016; Lauvaux et al., 2016).

*P8, L6: Please be more quantitative. It is not clear what has been reproduced. C3*

We have changed the text to generalize that the model sees enhancements downwind from major roadways and introduced a caveat that better details the limitations regarding model resolution.

The following text was added to ~p8L21:

The model generally produced mole fraction enhancements ($\Delta CO_2$) for grid cells containing or downwind from major roadways (Fig. 7). However, modeling intersection scale enhancements would require finer grid spacing capable of resolving sub-city-block spatial scales that is not yet feasible given current constraints on inventories, meteorological data, and computing resources.

*P8, L10 - 15: The simulated mole fractions are a combined result of transport and surface flux emissions. The authors, as mentioned, need to say how much we know about the surface emissions (used here) related to this discrepancy as well as the transport arguably improved from this work.*

See comments relating to p8L1.